B-box containing protein 1 from Malus domestica (MdBBX1) is involved in the abiotic stress response

Dai Yaqing 1
Lu Ying 1 2
Zhou Zhou 1
Wang Xiaoyun 1
Ge Hongjuan 3 ghj042@163.com
Sun Qinghua 1 qhsun@sdau.edu.cn
1 College of Life Science, Shandong Agricultural University , Taian, Shandong , China
2 Institute of Shandong River Wetlands , Jinan, Shandong , China
3 Qingdao Academy of Agricultural Science , Qingdao, Shandong , China
Silva Pedro
Electronic publication date: 2022 Feb 1
Publication date: 2022
Volume: 10
Electronic Location ID: e12852
Received 2021 Sep 22; Accepted 2022 Jan 7
Copyright: © 2022 Dai et al.
Copyright year: 2022
Copyright holder: Dai et al.
License: This is an open access article distributed under the terms of the Creative Commons Attribution License, which permits unrestricted use, distribution, reproduction and adaptation in any medium and for any purpose provided that it is properly attributed. For attribution, the original author(s), title, publication source (PeerJ) and either DOI or URL of the article must be cited.
License URL: https://creativecommons.org/licenses/by/4.0/

Keywords: MdBBX1, Malus domestica, Arabidopsis, Abiotic stress, ROS

Funding: National Natural Science Foundation of China 31872042 and 31972358 Natural Science Foundation of Shandong Province, China ZR2019MC040 and ZR2018MC022 Shandong Provincial Key Research and Development Project 2019JZZY010727 This work was supported by the National Natural Science Foundation of China (No. 31872042 and 31972358), the Natural Science Foundation of Shandong Province, China (No. ZR2019MC040 and ZR2018MC022), and the Shandong Provincial Key Research and Development Project (No. 2019JZZY010727). The funders had no role in study design, data collection and analysis, decision to publish, or preparation of the manuscript.

==============================
B-box proteins (BBXs), which act as transcription factors, mainly regulate photomorphogenesis. However, the molecular functions underlying the activity of plant BBXs in response to abiotic stress remain largely unclear. In this investigation, we found that a BBX from Malus domestica (MdBBX1) was involved in the response to various abiotic stresses. The expression of MdBBX1 was significantly upregulated in response to abiotic stresses and abscisic acid (ABA). Recombinant MdBBX1 increased stress tolerance in Escherichia coli cells. In addition, overexpression of MdBBX1 in Arabidopsis decreased sensitivity to exogenous ABA, resulting in a germination rate and root length that were greater and longer, respectively, than those of wild-type (WT) plants. Moreover, the expression of ABI5 was decreased in MdBBX1-overexpressing lines under ABA treatment. After salt and drought treatments, compared with the WT plants, the MdBBX1 transgenic plants displayed enhanced tolerance and had a higher survival rate. Furthermore, under salt stress, increased proline (PRO) contents, decreased levels of malondialdehyde (MDA), increased activity of antioxidant enzymes (superoxide dismutase (SOD), peroxidase (POD), catalase (CAT) and ascorbate peroxidase (APX)) and decreased accumulation of reactive oxygen species (ROS) were observed in the MdBBX1-overexpressing plants. Overall, our results provide evidence that MdBBX1 might play a critical role in the regulation of abiotic stress tolerance by reducing the generation of ROS.

Introduction

B-box containing proteins (BBXs) are typical zinc finger transcription factors with 1 or 2 zinc-binding B-box domain(s) at the N-terminus of protein sequence and occasionally with a CCT domain (for CONSTANS, CONSTANS-like, TOC1) at the C-terminus (Gangappa & Botto, 2014; Khanna et al., 2009). There are 32 family members of BBX in Arabidopsis thaliana and they are divided into five subfamilies according to their amino acid sequences. BBXs in group I-III contain a CCT domain that participates in the regulation of the transcription or nuclear import. Groups I, II, and IV contain two B-box motifs, while groups III and V harbor only one B-box motif (Gangappa & Botto, 2014; Khanna et al., 2009). BBX members in different groups have been identified to function in regulating anthocyanin accumulation, flowering, shade avoidance and photomorphogenesis, as well as responses to stress (An et al., 2019; An et al., 2020; Chang et al., 2008; Crocco et al., 2011; Crocco et al., 2015; Datta et al., 2007; Datta et al., 2008; Fang et al., 2019; Gangappa et al., 2013; Heng et al., 2019a; Heng et al., 2019b; Sarmiento, 2013; Wang et al., 2013; Wei et al., 2016; Yadav et al., 2019; Zhang et al., 2017).

In the BBX family of Arabidopsis, the members in the same group may have different functions. For example, both BBX21 and BBX24 belong to the same structural group, group IV (whose members have two B-box motifs and no CCT domain), but their functions in regulating photomorphogenesis are opposite. BBX21 is a positive regulator of photomorphogenesis, whereas BBX24 is a negative regulator (Xu et al., 2016). Despite the opposite functions of BBX21 and BBX24, the antagonistic modulating ability of both depends on HY5 (Job et al., 2018), which is a central downstream regulator of light-mediated developmental processes and can bind to the promoter of the ABI5 gene to activate its expression (Chen et al., 2008). To illustrate the molecular mechanism for underlying the contrasting functions of BBX21 and BBX24 (Job et al., 2018), the protein sequences of these two genes were compared, and the results revealed that their functional differences were mainly determined by different sequences of the C-terminal region. In support of this notion, the researchers constructed two vectors that expressed BBX24 and BBX21 proteins fused to each other’s C-terminal sequences; the fusion proteins were subsequently named “BB24C21” and “BB21C24”, respectively. The results showed that, similar to the overexpression of BBX21, the overexpression of BB24C21 could transcriptionally upregulate the expression of HY5, whereas over-expression of BB21C24 did not have any effect on the mRNA levels of HY5 (Job et al., 2018). Furthermore, the researchers found that BBX21 could mediate HY5 post transcriptionally. In contrast, BBX24 could prevent HY5 from binding to the promoter of the target gene, probably by heterodimerizing with HY5 and inhibiting its ability to bind to DNA. In conclusion, closely related BBXs may perform opposite functions.

BBXs are also involved in the stress response (Gangappa & Botto, 2014). Act as a kind of salt tolerance-related protein, BBX24 can negatively regulate the expression of many stress-related genes (Nagaoka & Takano, 2003). AtBBX24 transgenic plants were shown to be more salt tolerant than wild-type (WT) plants under salt stress. BBX5, a group I member, contains two B-box domains and one CCT domain, and is involved in the response to abiotic stress through the abscisic acid (ABA)-dependent signaling pathway. Overexpressing BBX5 can notably improve the plant resistance to abiotic stresses (Min et al., 2015). Overexpression of a BBX protein in banana obviously improved its tolerance to biotic and abiotic stresses, such as pathogen infection and chilling (Chen et al., 2012). Similarly, overexpression of OsBBX25 in Arabidopsis thaliana can increase plant tolerance to abiotic stresses (Liu et al., 2012). Heterologous expression of AtBBX21 enhances the photosynthesis rate and alleviates photoinhibition in Solanum tuberosum (Crocco et al., 2018). In addition, some tomato BBX genes can also be induced in response to heat, drought and phytohormones (Chu et al., 2016). Overall, BBX proteins play vital roles in regulating various stress responses.

Our previous study reported that there are 64 BBXs in the apple genome, which can be divided into five groups, similar to the Arabidopsis BBX family. Some MdBBX genes are induced in response to different abiotic stresses, indicating that MdBBXs may participate in abiotic stress responses (Liu et al., 2018). A recent study found that MdBBX10 from apple could promote tolerance to drought and salt stresses in Arabidopsis (Liu et al., 2019a). MdBBX10 belongs to group V, and contains one B-box domain, but no CCT domain. Overexpression of MdBBX10 in Arabidopsis dramatically improved the tolerance to abiotic stress and increased sensitivity to ABA during the seed germination and seedling stages (Liu et al., 2019a). Here, we demonstrated that a BBX member of group I, MdBBX1, which contains two B-box domains and a CCT domain (Fig. S1), also responds to abiotic stress, but causes insensitivity to exogenous ABA when overexpression in Arabidopsis, which is opposite to the response of MdBBX10 to ABA.

Materials and Methods

Plant growth conditions and treatments

For organ-specific expression analyses, different apple organs, including roots, stems, leaves, flowers and fruits, were sampled from 6-year-old apple trees growing at the experimental station of Shandong Agricultural University (Tai’an, Shandong, China).

Apple (golden delicious) seedlings were cultivated under greenhouse conditions (relative humidity of 60–75%) at 22 ± 1 °C with a 16 h light/8 h dark photoperiod for approximately 1 year. Then, the uniformly growing seedlings were selected for stress treatments. For salt and drought stress treatments, the apple seedlings were watered with solutions of 250 mM NaCl or 25% (w/v) polyethylene glycol-6000 (PEG-6000), and control seedling received the same amount of water only. For ABA treatment, 100 µM ABA solutions were directly sprayed on the seedlings. For cold stress, the apple seedlings were subjected to 4 °C condition, while seedlings growing at room temperature (25 °C) were used as the controls. Samples were collected from three different kinds of seedlings at 0, 3, 6, 9 and 12 h after treatment, as was done in a previous study (Yuan et al., 2013). Then, the collected samples were immediately frozen in liquid nitrogen and stored at −70 °C till used. Subsequently, the total RNA was extracted from the collected samples using an improved cetyl-trimethylammonium bromide (CTAB) procedure (Gasic, Hernandez & Korban, 2004).

The seeds of WT (Col-0) and transgenic Arabidopsis were disinfected and sown on 1/2 Murashige and Skoog (MS) media. After culturing at 4 °C for 3 days to undergo vernalization, the seedlings were transferred to a greenhouse condition, which included a 22 ± 1 °C temperature with a 16 h light/8 h photoperiod. Then, 3-week-old WT and transgenic plants were treated with 250 mM NaCl or 25% (w/v) PEG-6000 as described by Liu et al. (2019a), and the control seedlings were treated with water only. Plant growth status was observed and the survival rates were determined daily. Each treatment was performed at least three times.

Quantitative real-time PCR (qRT-PCR) analysis

QRT-PCR is a commonly used approach for the quantitative detection of gene expression in real time (Deepak et al., 2007). All the primers used in this investigation were designed according to the target gene sequences via the Beacon Designers software and were shown in Table S1. qRT-PCR was carried out using a SYBR® PrimeScript™ RT-PCR Kit (TaKaRa, Dalian, China) and run on a CFX96TM Real-Time PCR Detection System (Bio-Rad, Hercules, CA, USA). The Arabidopsis Actin8 and apple actin genes were used as reference genes (Wang et al., 2016). The thermal cycling parameters were as follows: 40 cycles of 95 °C denaturation for 15 s, 55 °C annealing for 20 s and 70 °C extension for 15 s. The qRT-PCR data were analyzed by the 2−ΔΔCt method (Livak & Schmittgen, 2001). The relative expression of MdBBX1 in the treated samples was compared with that in the nontreated samples at each treatment time point with significant differences (P < 0.05) determined based on Tukey’s multiple test.

Subcellular localization of MdBBX1

The full-length coding sequence of MdBBX1 was amplified from apple and inserted into the pROKII vector containing a GFP gene and the CaMV35S promoter. Cells of Agrobacterium tumefaciens GV3101 with the recombinant plasmid were cultured overnight, resuspended in osmotic solution (10 mM, Mole MgCl2, 10 mM 2-[N-morpholino] ethanesulfonic acid (MES) and 150 mM acetosyringone), and then injected into the leaves of 1-month-old Nicotiana benthamiana plants. The fluorescent signal of MdBBX1-GFP was detected via a confocal microscope (LSM 510 META, Carl Zeiss, Jena, Germany) after 2–3 days. The nuclei were subsequently stained with 100 g/mL 4′, 6 -diamidino-2-phenylindole (DAPI) (Solarbio, Beijing, China) for 10 min. Leaves overexpressing 35S-GFP were used as controls (Wang et al., 2018).

Construction of expression plasmids

The cDNA sequence of MdBBX1 was inserted into the polyclonal sites of pET-30a (+) (Novagen), which contained His-tagged sequences. Then, the recombinant vector was transformed into Escherichia coli BL21 cells. The recombinant sequences in the plasmids were sequenced by Sangon Biotechnology Company (Shanghai, China).

Survival test of Escherichia under different abiotic stresses

Survival analysis of Escherichia coli under salt and drought stress was conducted as described by Du et al. (2014). The cells were cultured in Luria-Bertani (LB) liquid media until the OD600 reached 0.4–0.6, and then the expression of the recombinant protein was induced for 2 h using isopropyl β-D-1-thiogalactopyranoside (IPTG) at 37 °C. All the bacterial cultures were first diluted to an OD600 of 0.6 and then diluted 10−3, 10−4 and 10−5 times. For the survival test on solid media, 10 µL cultures of each dilution were spotted onto solid LB media that included 500 mM KCl, 500 mM NaCl or 600 mM mannitol and incubated for 12 h at 37 °C. Then, the colony numbers in each dish for the culture diluted to 10−5 were counted. Each experiment was performed at least three times.

For the survival test in liquid media, the cultures were first diluted to an OD600 of 0.6, after which 200 µL of the cultures were put into 20 mL of LB solution that included 500 mM NaCl, 500 mM KCl or 600 mM mannitol and incubated at 37 °C on a rotary shaker (150 rpm). Then, the bacterial suspension was collected every 2 h for 24 h, after which the OD600 of the culture was measured. Each experiment was repeated at least 3 times.

Generation of transgenic plants

The pROKII-MdBBX1 recombinant plasmids were transformed into Arabidopsis in accordance with the floral-dip method via Agrobacterium tumefaciens (GV3101)-mediated transformation. Subsequently, the MdBBX1 overexpression seedlings were screened on MS agar media supplemented with 50 µg/mL kanamycin and were further identified via PCR using MdBBX1 and GFP primers. The specific primers used are shown in the Table S1.

Analysis of germination status under different abiotic stresses

Fifty seeds of WT or overexpression (OE) lines were sown onto 1/2-strength MS agar media supplemented with different concentrations of mannitol (300 or 400 mM), NaCl (150 or 200 mM) or ABA (0.2 or 0.6 µM). Seed germination was observed and measured every 12 h. For root length analysis, the seeds were grown vertically on 1/2-strength MS media as described above. The root length of 20 seedlings was measured after 10 days, and each treatment was performed at least three times.

Measurements of proline (PRO), malondialdehyde (MDA), and reactive oxygen species (ROS) contents and antioxidant enzyme activity

For physiological index measurements, the free PRO content was measured using a spectrophotometric PRO kit (Solarbio Life Sciences, Beijing, China). The MDA content was measured using a thiobarbituric acid reactive substances assay (Aguilar Díaz de León & Borges, 2020; Hodges, Delong & Prange, 1999) and the contents of hydrogen peroxide (H2O2) and superoxide anions (O2.−) were measured using O2.− and H2O2 kits, respectively (Nanjing Jiancheng Bioengineering Institute, China). Similarly, the total protein contents were determined using a BCA Protein Assay Kit (Nanjing Jiancheng Bioengineering Institute, Nanjing, China), and the activities of superoxide dismutase (SOD), peroxidase (POD), catalase (CAT) and ascorbate peroxidase (APX) were measured based on the protocols of the corresponding kits (Nanjing Jiancheng Bioengineering Institute, Nanjing, China) (Bai et al., 2020; Li et al., 2019; Ma et al., 2019).

Statistical analysis

All experiments were conducted at least three times. The data presented are the means ± standard deviations of three replications. Statistical significance was analyzed using SPSS software (version 17.0), and Turkey’s multiple range comparison tests were performed to determine the significance of differences between samples (P < 0.05 or P < 0.01).

Results

Organ-specific expression pattern analysis and subcellular localization of MdBBX1

The organ-specific expression pattern of MdBBX1 was analyzed via qRT-PCR. The results revealed that the transcript level of MdBBX1 was obviously higher in the leaves than in other organs (Fig. 1A). To identify the subcellular localization of MdBBX1, an MdBBX1-GFP construct and empty GFP plasmid were introduced into epidermal cells of tobacco leaves, after which the nuclei were stained by DAPI. The fluorescence signal and DAPI staining were predominantly distributed in the nucleus (Fig. 1B), which indicated that MdBBX1 was localized there.

Figure 1 Expressional pattern and subcellular localization analysis of MdBBX1.

(A) qRT-PCR analysis of MdBBX1 expression in different organs of 6-year old apple seedlings. The experiments were repeated three times and vertical bars indicate the standard error of the mean. The letters above the columns represent significant differences (P < 0.05) based on Tukey’s multiple test. (B) Subcellular localization of MdBBX1-GFP fusion protein. 35S::MdBBX1-GFP construct was transformed into tobacco leaves and was examined in the epidermal cells at 48 h after the transformation by confocal fluorescence microscopy. The nuclei were stained with 100 g/mL DAPI for 10 min.

The MdBBX1 promoter contains elements related to the abiotic stress response

Some BBX members were found to respond positively to abiotic stresses in a previous study (An et al., 2020; Crocco & Botto, 2013; Liu et al., 2018; Shalmani et al., 2018). To explore the potential functions of MdBBX1 in response to a variety of abiotic stresses, the DNA sequence within 2000 bp upstream of MdBBX1 (the promoter sequence) was scanned via PlantCARE software (http://bioinformatics.psb.ugent.be/webtools/plantcare/html/). The results showed that many cis-acting elements that may be involved in responses to abiotic stress, light and other signals were present in the MdBBX1 promoter region (Table 1). For instance, MBS (MYB-binding site) elements are involved in the response to drought stress, and ABA-responsive elements (ABREs) function in response to exogenous ABA. LTRs are found to participate in low-temperature responsiveness. In addition, some cis-acting elements such as TC-rich repeats and TCA elements are involved in responses to defense and stress or to salicylic acid. Taken together, these results indicated that MdBBX1 may participate in the response to abiotic stress.

Table 1 Putative cis-acting elements of the promoter of MdBBX1.

Cis-element	Position	Sequence(5′-3′)	Function	
ABRE	−1331	ACGTG	cis-acting element involved in the abscisic acid responsiveness	
CGTCA-motif	−1794	CGTCA	cis-acting regulatory element involved in the MeJA-responsiveness	
G-Box	−1330	CACGTT	cis-acting regulatory element involved in light responsiveness	
GT1-motif	−1523	GGTTAAT	light responsive element	
LTR	−1992	CCGAAA	cis-acting element involved in low-temperature responsiveness	
MBS	−577	CAACTG	MYB binding site involved in drought-inducibility	
P-box	−1778	CCTTTTG	gibberellin-responsive element	
TC-rich repeats	−941	ATTCTCTAAC	cis-acting element involved in defense and stress responsiveness	
TCA-element	−1242	CCATCTTTTT	cis-acting element involved in salicylic acid responsiveness	

To further investigate whether MdBBX1 is expressed in response to abiotic stress, apple seedlings were treated with solutions of 100 µM ABA, 25% polyethylene glycol (PEG), or 250 mM NaCl or 4 °C for different durations. As shown in Fig. 2, the transcript levels of MdBBX1 were upregulated in both the leaves and the roots under exogenous ABA, salt and PEG treatment, MdBBX1 expression was maximized after 6 h of stimulation by ABA, PEG or NaCl compared with the control in the roots and increased by approximately 120-, 4- and 2-fold, respectively. In addition, the transcript levels of MdBBX1 were upregulated by 12-, 20- and 10-fold after treatment with ABA, PEG or NaCl in the leaves. Notably, the expression of MdBBX1 was significantly upregulated under cold conditions only in the roots. Taken together, the above results showed that the expression of MdBBX1 was induced in response to different abiotic stresses, which suggested that MdBBX1 may be involved in the response to abiotic stress.

Figure 2 The expression pattern of MdBBX1 in (A) roots and (B) leaves under various abiotic stresses.

Each column represents the mean values of three biological replicates and vertical bars indicate the standard error of the mean. The letters above the columns represent significant differences (P < 0.05) based on Tukey’s multiple test.

Ectopic expression of MdBBX1 in Escherichia improved cell tolerance to abiotic stress

To determine the stress resistance function of MdBBX1, heterogeneous expression of MdBBX1 was induced in Escherichia coli growing on solid media under different stress conditions. Survival tests of Escherichia coli cells were carried out, with empty vectors used as controls. As shown in Figs. 3A–3B, the growth of the cells in nonstress media showed slight significant difference between those harboring MdBBX1 and those harboring the empty vector. However, when the same concentration of cultures was inoculated onto plates with stress media, the number of Escherichia coli colonies expressing MdBBX1 was significantly higher than that of the control colonies harboring the empty vector. To further confirm the function of MdBBX1, a growth curve of Escherichia coli in liquid media was constructed. As shown in Fig. 3C, under nonstress conditions, few differences were observed among the growth curves of cells with and without MdBBX1 expression. However, under different stress conditions, the growth rate of Escherichia coli expressing MdBBX1 was significantly faster than that of the control cells carrying the empty vector. The results suggested that MdBBX1 provided strong abiotic stress tolerance.

Figure 3 Survival test of E. coli cells carrying MdBBX1 or empty vector under various stress conditions.

(A) A total of 10 μL cultures induced by 1 mM IPTG for 2 h (OD600 about 0.5) were diluted from 10−3 to 10−5 and were spotted on solid medium containing NaCl, KCl or mannitol. Each experiment was carried out in three biological replicates. (B) The colony numbers appearing on above medium were counted in 10−5 concentration. Mean values are from three independent replicates and error bars indicate standard deviation. The letters above the columns represent significant differences (P < 0.05) based on Tukey’s multiple test. (C) Growth curve of E. coli cells in LB liquid medium with and without addition of 1 mM IPTG under NaCl, KCl or mannitol treatments. The mean expression value was calculated from three independent replicates. Vertical bars indicate the standard error of mean, ** and * indicate significant differences compared with vector cells at P < 0.01 and P < 0.05, respectively.

Overexpression of MdBBX1 increased resistance to abiotic stress in Arabidopsis

To determine the role of MdBBX1 in abiotic stress resistance in plants, three transgenic lines (OE1, OE2 and OE3) with similar expression levels of MdBBX1 were obtained and subjected to salt and drought treatment (Fig. S1). As shown in Fig. 4A, on normal media, the germination rate and growth status of seedlings exhibited no obvious differences between the WT and transgenic lines. However, under salt stress, the OE seeds presented a significantly higher germination rate than did the WT seeds. On the stress media that included 200 mM NaCl, the germination rate of the OE seeds reached 60% compared with 20% for WT seeds after treatment for 48 h. In addition, the root length of OE lines was obviously longer than that of the WT plants on 150 mM NaCl media (Fig. 4B). When the 3-week-old seedlings were watered with 250 mM NaCl for 10 days, the leaves of WT began to turn yellow, but few yellow leaves were observed on the OE seedlings. After treatment for 15 days, more wilted and chlorotic leaves were observed on the WT than in the OE lines. After treatment with salt for 20 days, the OE plants presented a significantly higher survival rate (approximately 90%) than did the WT plants (approximately 40%) (Fig. 5A). In addition, under salt stress, the PRO accumulation in the OE plants was obviously higher than that in the WT (Fig. 5B), while the levels of MDA were obviously lower in the OE lines than in the WT (Fig. 5C).

Figure 4 Germination and root length phenotypes of MdBBX1 overexpression plants under salt tolerance.

(A) Germination phenotype of the WT and MdBBX1 -overexpressed (OE) lines on 1/2 MS medium containing NaCl (0, 150 and 200 mM). Three independent experiments were conducted and each phenotype included 50 seeds. The mean expression value was calculated from three independent replicates. Vertical bars indicate the standard error of mean, **P < 0.01 and *P < 0.05 compasred with WT. (B) The root length of WT and MdBBX1-transgenic lines in 1/2 MS medium containing 150 mM NaCl. Root growth was measured after NaCl treatment for 14 days. The letters above the columns represent significant differences (P < 0.05) based on Tukey’s multiple test.

Figure 5 Overexpression MdBBX1 enhanced salt tolerance in transgenic plants.

(A) The representative phenotypes of 3-week old WT and OE seedlings were treated with NaCl (250 mM) for 3 days to 20 days. (B) Survival rates of WT and transgenic plants after salt stress. PRO (C) and MDA content (D) were measured in WT and transgenic plants after salt stress. Mean values are from three independent replicates and error bars indicate standard deviation. The letters above the columns represent significant differences (P < 0.05) based on Tukey’s multiple test.

Similarly, after mannitol treatment, the OE lines also displayed significantly higher germination rates and longer root lengths than did the WT line (Figs. 6A–6B). Moreover, the transgenic plants presented a higher survival rate than did the WT plants when treated with 25% PEG-6000 (Fig. 6C). When treated for 25 days, nearly 60% of WT plants wilted and died, while the leaves of OE plants yellowed slightly. The survival rate of the OE plants was approximately 90%, which was significantly higher than the survival rate of the WT plants. Taken together, these results indicated that overexpression of MdBBX1 enhanced the abiotic stress resistance of the transgenic plants during germination and vegetative stage.

Figure 6 The phenotype of WT and MdBBX1 transgenic plants in response to drought stress.

(A) The seed germination of WT and the MdBBX1 transgenic plants on 1/2 MS medium containing mannitol (0, 300, or 400 mM). Three independent experiments were conducted and each phenotype included 50 seeds. Vertical bars indicate the standard error of mean, ** and * indicate significant differences in comparison with WT at P < 0.01 and P < 0.05, respectively. (B) The root length of WT and the transgenic plants of MdBBX1 in 1/2 MS medium containing 300 mM mannitol. Root growth of WT and the transgenic plants of MdBBX1 was measured after 14 days. (C) The representative phenotypes of WT and OE seedlings were treated with 25% (w/v) PEG-6000 for 10 days and 25 days. All the columns were represented as mean values of three independent replicates and error bars indicate standard deviation. The letters above the columns represent significant differences (P < 0.05) based on Tukey’s multiple test.

Overexpression of MdBBX1 in Arabidopsis decreased sensitivity to ABA

ABA plays a critical role in the physiological regulation of plant development in seed germination and in abiotic stress responses (Vishwakarma et al., 2017). To determine the potential function of MdBBX1 in response to ABA, seeds of the OE and WT lines were plated on MS media either without or with ABA. Under MS media without ABA, the WT and OE lines displayed similar germination rates. However, under ABA treatment, the germination rates of the OE lines were significantly higher than those of the WT (Fig. 7A). Moreover, after the seedlings grew on MS media with ABA for 10 days, the primary root length of the OE seedlings was obviously greater than that of the WT seedlings (Fig. 7B). To determine the effect on root growth, the seeds of WT and OE were sown on MS media for 2 days first and then transferred to plates with media that included 0.6 μM ABA. The root length of the OE seedlings was still longer than that of the WT seedlings (Fig. S2). The above results suggested that overexpression of MdBBX1 decreased ABA sensitivity in Arabidopsis during the germination and seedling stages.

Figure 7 Germination phenotype of WT and transgenic plants in response to ABA.

(A) Seed germination of WT and OE lines on 1/2 MS medium containing different ABA concentrations (0, 0.2 or 0.6 µM). Three independent experiments were conducted and each phenotype included 50 seeds. Vertical bars indicate the standard error of mean, ** indicate significant differences in comparison with WT at P < 0.01. (B) The root length of WT and OE lines on 1/2 MS medium containing 0.6 µM ABA. Root growth of WT and OE seedlings was measured after 14 days. (C) The transcriptional levels of HY5 and ABI5 in WT and OE lines after 100 µM ABA treatment for 3 h. Little difference was observed among different lines, so one of the OE MdBBX1 transgenic lines was selected as the representative for gene expression analysis. The letters above the columns represent significant differences (P < 0.05) based on Tukey’s multiple test.

In the ABA signaling pathway, HY5 and ABI5 are crucial for seed germination and seedling development (Chen & Xiong, 2008; Finkelstein, Gampala & Rock, 2002; Finkelstein & Lynch, 2000b). To determine whether the development of MdBBX1-overexpressing seedlings under abiotic stresses was related to HY5 or ABI5, the transcript levels of ABI5 and HY5 were measured after ABA treatment. The results revealed that the expression of ABI5 was markedly reduced in the OE plants compared with the WT plants, while HY5 changed only slightly (Fig. 7C). Another OE line overexpressing a different MdBBX family member (MdBBX10), which is ABA sensitive, was also evaluated under the same treatment (Liu et al., 2018). As shown in Fig. 7C, the changes in ABI5 expression levels were opposite between MdBBX1 and MdBBX10, which was reasonably expected in terms of a response to ABA treatment.

Overexpression of MdBBX1 reduced ROS accumulation in transgenic plants

Various abiotic stresses often lead to the accumulation of excessive amounts of ROS, particularly H2O2 and O2•−, which has an important impact on plant growth and development (Mittler et al., 2004). To analyze whether MdBBX1 responds to abiotic stress through the regulation of ROS levels, the accumulation of O2•− in WT and OE plants was assessed via nitro blue tetrazolium (NBT) staining. No obvious difference was found between the WT and OE lines. However, under NaCl and PEG conditions, the OE lines accumulated lower levels of O2•− than the WT did (Fig. 8A). Furthermore, the contents of H2O2 and O2•− were measured, and the results showed that their contents in the OE lines were significantly lower than those in the WT (Figs. 8B, 8C). Similarly, under normal conditions, the activities of SOD, POD and APX were not notably different between the OE and WT lines. However, after salt treatment, the activities of SOD, POD and APX in the OE lines significantly increased compared with those in the WT lines (Fig. 9). Together, these results suggested that overexpression of MdBBX1 could decrease the accumulation of ROS in transgenic plants by mediating the activities of ROS-scavenging enzymes.

Figure 8 Analysis of ROS in WT and the MdBBX1 transgenic plants after salt and drought stress.

(A) NBT staining of O2•− in WT and OE plants after the treatment with NaCl or PEG-6000 for 6 h. (B) The content of H2O2 and O2•− in WT and transgenic lines after NaCl treatment. Each column represents the average of three replicates and error bars indicate standard deviation. The letters above the columns represent significant differences (P < 0.05) based on Tukey’s multiple test.

Figure 9 The activities of ROS-scavenging enzymes in WT and the MdBBX1 transgenic plants after NaCl treatment.

Each column represents the average of three replicates and error bars indicate standard deviation. The letters above the columns represent significant differences (P < 0.05) based on Tukey’s multiple test.

Discussion

Many BBXs are involved in the response to abiotic stresses in plants. For example, AtBBX5 and AtBBX24 are positive regulators that modulate the drought and salt stress resistance in Arabidopsis (Min et al., 2015; Nagaoka & Takano, 2003). Overexpression of OsBBX25 in Arabidopsis increased the tolerance to abiotic stresses (Liu et al., 2012). Similarly, heterologous, constitutive expression of CmBBX22 in Arabidopsis reduced seed germination and seedling growth under exogenous ABA, but improved plant drought tolerance (Liu et al., 2019b). A recent study found that overexpression of MdBBX10 in Arabidopsis can promote the salt and drought tolerance, and the transgenic seedlings were shown to hypersensitive to exogenous ABA (Liu et al., 2019a). In this investigation, overexpression of MdBBX1 also enhanced tolerance to abiotic stresses. However, the phenotype of MdBBX1 overexpression plants was different from that of MdBBX10 transgenic plants under exogenous ABA treatment. MdBBX10 overexpressing plants were hypersensitive to exogenous ABA, while the MdBBX1 transgenic plants were insensitive to ABA.

As a pivotal phytohormone, ABA is extensively involved in the regulation of plant growth and development (Wang et al., 2019), especially in response to various abiotic stresses and seed germination (Finkelstein, Gampala & Rock, 2002). During the initial stages of germination, the endogenous ABA content in seeds decreases rapidly and markedly after imbibition (Ali-Rachedi et al., 2004; Gubler, Millar & Jacobsen, 2005; Jacobsen et al., 2002). When exogenous ABA is added, seed germination and seedling growth can be repressed (Finkelstein, Gampala & Rock, 2002; Finkelstein & Lynch, 2000a). ABA-insensitive genes (ABIs), especially ABI5, play a vital role in ABA signaling and photomorphogenesis. ABI5 is mainly expressed in dry seeds, and is involved in ABA-dependent growth arrest when seed dormancy is broken (Finkelstein, Gampala & Rock, 2002; Finkelstein & Lynch, 2000b). The efficiency of the ABA-dependent growth arrest is directly dependent on ABI5 levels (Brocard, Lynch & Finkelstein, 2002; Lopez-Molina, Mongrand & Chua, 2001). ABI5 markedly decreases after germination but can be induced by exogenous ABA (Finkelstein & Lynch, 2000b; Lopez-Molina, Mongrand & Chua, 2001). Moreover, the expression of ABI5 can be activated by HY5 through direct binding to its promoter in Arabidopsis (Chen et al., 2008). BBX family members also regulate the expression of ABI5. For example, BBX19 from Arabidopsis suppresses seed germination by inducing expression of ABI5 (Bai et al., 2019). However, in this investigation, the transcript level of ABI5 significantly decreased in the MdBBX1 OE plants compared with the WT plants, whereas it was significantly increased in the MdBBX10 overexpression plants (Fig. 7C). Moreover, the transcript levels of HY5 did not obviously change in the MdBBX1 OE seedlings compared with the WT seedlings, but they were obliviously lower than those in the MdBBX10 OE seedlings. These results are consistent with the phenotypes during seed germination after ABA treatment. In addition, the results of multiple sequence alignment revealed 17%, 36%, and 16% homology between HY5 and BBX1, BBX5, and BBX21, respectively (Fig. S3), which suggested that overexpression of MdBBX1 may have little effect on the expression of endogenous genes. Taken together, these results suggested that MdBBX1 may interfere with the expression of ABI5 and HY5 to promote seed germination and seeding growth in transgenic plants.

Abiotic stress can disrupt the normal homeostasis of plants, leading to the production of ROS, mainly comprises of H2O2 and O2•− (Miller et al., 2010). Low concentrations of ROS act as critical signaling molecules that are beneficial to plant growth and development, especially when plants are exposed to extreme environmental conditions (Schippers et al., 2012). However, the accumulation of excessive amounts of ROS leads to very serious oxidative damage to plant cells and represses the normal growth of plants (Mullineaux & Baker, 2010). To alleviate oxidative damage, plant cells immediately employ a series of response mechanisms to suppress ROS production, such as the activation of ROS-scavenging enzymes (SOD, POD, CAT and APX) (Miller et al., 2010). Abiotic stress often increases the accumulation of ROS, thus causing membrane damage with lipid peroxidation, generating MDA. Our results suggested that the ROS and MDA contents were no significant different between the OE and WT lines under normal conditions. However, under salt treatment, compared with the WT line, the MdBBX1 transgenic lines displayed a greater ROS-scavenging ability and antioxidant enzyme activities (Fig. 9), and a lower ROS accumulation (Fig. 8) and MDA levels (Fig. 5). Regulation of antioxidant capacity through improving the ROS-scavenging system might be a common mechanism to increase salt tolerance. Similar to our study, a previous study reported that overexpression of ThSOS3 from Tamarix hispida improved the salt tolerance of transgenic plants by alleviating the accumulation of ROS, decreasing the accumulation of MDA accumulation and increasing the activity of antioxidant enzymes (Liu et al., 2020). In addition, overexpression of apple MdMIPS1 also enhanced salt tolerance by increasing the activities of SOD, POD, and decreasing ROS and MDA contents in transgenic apple under salt stress (Hu et al., 2020). Taken together, these results indicated that MdBBX1 provides salt stress resistance by enhancing ROS-scavenging system and alleviating oxidative stress.

Conclusions

The transcript level of MdBBX1 increased in response to various stresses. Overexpressing MdBBX1 in Arabidopsis improved abiotic stress tolerance by regulating ABA signaling and the production of ROS. However, the detailed molecular mechanisms underlying these phenomena still need to be tested in future experiments.

Supplemental Information

Supplemental Information 1 Supplementary Table and Figures.

Click here for additional data file.

Supplemental Information 2 Identification of the transgenic plants of MdBBX1.

Phenotype of OE1, OE2 and OE3 plants; B. PCR products of transgenic plants; C. The expression of MdBBX1 in the leaves of WT and transgenic plants.

Click here for additional data file.

Supplemental Information 3 Roots length of WT and OE seedlings in medium containing ABA.

The wild type and OE seedlings were grown on MS for 2 days and then were transferred to 1/2 MS medium containing 0.6 µm ABA.

Click here for additional data file.

Supplemental Information 4 The homology among MdBBX1, AtBBX1, AtBBX5 and AtBBX21.

Only 17%, 36%, 16% homology between MdBBX1 and AtBBX1, AtBBX5, and AtBBX21, respectively.

Click here for additional data file.

Additional Information and Declarations

Competing Interests

Author Contributions

Data Availability

The authors declare that they have no competing interests.

Yaqing Dai conceived and designed the experiments, performed the experiments, analyzed the data, prepared figures and/or tables, and approved the final draft.

Ying Lu conceived and designed the experiments, analyzed the data, prepared figures and/or tables, authored or reviewed drafts of the paper, and approved the final draft.

Zhou Zhou performed the experiments, analyzed the data, prepared figures and/or tables, and approved the final draft.

Xiaoyun Wang conceived and designed the experiments, authored or reviewed drafts of the paper, and approved the final draft.

Hongjuan Ge analyzed the data, authored or reviewed drafts of the paper, and approved the final draft.

Qinghua Sun conceived and designed the experiments, analyzed the data, authored or reviewed drafts of the paper, and approved the final draft.

The following information was supplied regarding data availability:

The data is available at figshare: zhou, zhou (2022): MdBBX1 raw data. figshare. Dataset. https://doi.org/10.6084/m9.figshare.16575233.v1.

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
