# Peer review of "B-box containing protein 1 from Malus domestica (MdBBX1) is involved in the abiotic stress response"

_PeerJ, doi:10.7717/peerj.12852_

## Round 0.1 · original submission · Major Revisions

Please address all issues highlighted by our reviewers. You should note that reviewer #1 has also uploaded a commented version of your submission, with additional remarks

Reviewer 1 ·

Basic reporting

The presented research describes the functional annotation of a BBX gene isolated from apple. Several experiments are performed to determine the gene-of-interest’s native expression in different tissue of apple seedlings, the protections it provided to transformed E. coli cells and the role it plays in transgenic plants overexpressing the GOI when exposed to abiotic stress. The presented data fall within the broad scope of general plant and molecular biology as required by PeerJ. The presented research conveys novel data regarding the function of an apple BBX TF gene.

The grammar used in the article is not good and the authors need to address all the linguistic aspects of the paper before resubmitting. I have indicated numerous mistakes and editorial issues on the attached pdf file. In general, the authors should proofread and make the necessary corrections across the whole document. Also, the authors should decide on one tense and stick to that.

The abstract reflects the content of the paper but the grammar in some instances need to be improved by the authors. The introduction section covers relevant and recent information on the presented topic.

The Material and Method section is adequately described but numerous mistakes need to be corrected as indicated in the attached pdf. Give complete information regarding kits used.

Sections of the Discussion are repeating the Results too extensively and should be shorten.

Experimental design

The results are presented in a systematic and relevant manner. The statistical analysis is sound. However, a few issues the authors need to address:
The authors should remove the Discussion sections from the Results, or change it all together to a Results and Discussion section.
Fig 1A, the authors are working with one year old apple seedlings (as per fig legend), does such small plants really have fruits to analyse?
Unclear how tobacco plants were transformed for subcellular localization studies, this is not explained in the M&M, and details are missing. In the Results it seems like tobacco epidermal leave cells were transformed. No controls are included for the subcellular localization results, it is likely the expression occur in the nucleus but without a control this cannot be concluded.
In Table 1, + and – positions for cis-elements are indicted but the authors said that the 1.5 kb upstream region of the gene was analyses, why then + and -?
Fig 2 – it would be better if the authors can standardise the y-axis of the graphs across treatments, at least in the graphs where the values are within the same range.
Why are the line graphs in Fig 3 in colour but not the line graphs in Fig 4?
Fig 4B graph – where is the stats for the Control treatment?
Why is the qPCR done with PEG but the seedlings germination/rooting tested with mannitol (Fig 6a,b) and seedling growth on PEG? Why the authors jumped between osmoticums are unclear.
Legend of Fig 7 do not correspond with the presented figures around ABA treatment.
Why did the authors only test one of the OE MbBBX1 transgenic lines for ABA-related gene expression (Fig 7)?
Figure 8 has a Fig 7 legend about ABA and images linked to ROS, this needs to be fixed. Similar confusion around legends for Fig 9.

Validity of the findings

Overall, experiments seem to be conducted in a scientifically sound manner and presented in a way that address the scientific aim of this paper. However, numerous mistakes and improvements need to be made to the paper before this data set can be considered for publication.

Annotated reviews are not available for download in order to protect the identity of reviewers who chose to remain anonymous.

·

Basic reporting

The manuscript entitled “B-box containing protein 1 from Malus domestica (MdBBX1) involved in abiotic stress response” submitted by Dai et al shows great effort in elucidating MdBBX1 might play an important role in the regulation of tolerance to abiotic stresses. However, the terms and content organization need to be rephrased to highlight importance of their work.

Experimental design

The experimental design is overall reasonable. Please state the evidence for the suitable choice of PEG and NaCl concentrations.

Validity of the findings

MdBBX1 might play an important role inthe regulation oftolerance to abiotic stresses by reducing the
generation of ROS. The result in this study is informative, and the findings are novel.

Additional comments

1. In terms of the main issue existing in this manuscript, the language should be improved.
2. Please carefully check the entire text and make sure the genes name should be in italic, but not the protein.
3. L24, remains should be remain.
4. L25-26, various abiotic stresses response should be stress responses.
5. L29, longer length should be rephrased.
6.L33, SOD, POD, AND APX should be summarized into antioxidant enzymes, noy peroxidases.
7. L104, what is the meaning of different time?
8. Line 129, “Su et al., 2013” can be replaced with “Livak and Schmittgen 2001” https://doi.org/10.1006/meth.2001.1262
9. Line 180-185, recently-published papers about POD, CAT, SOD and APX should be cited in the manuscript.
10. In the methodology sections, there is no information about the statistical analysis. Please add a separate section of “statistical data analysis”.
11. In figure 3, 4, and 7 legend, * should be indicate P< 0.05 usually, **, P < 0.01, ***, P < 0.001.
12. Please check figure 7, 8, 9. The figure does not match the legend.

---

## Round 0.2 · Minor Revisions

Most of the issues have been dealt with, but some minor changes are still required. Please address them

Reviewer 1 ·

Basic reporting

The authors improved the paper significantly. A few remaining gramma mistakes are indicated in the attached pdf.

Experimental design

A well constructed and executed set of experiments.

Validity of the findings

Valet data set.

Additional comments

Fig 1 – why not just use the Supplementary figure that show the DAPI control in the paper?
Fig 3B – The significant different indicator “b” at the control and Nacl treatment must be a mistake, its highly unlikely that the two samples do not differ significantly
Fig4B – The significant difference indicated (a) is highly unlikely. Does the transgenic lines under control treatment really not differ from the NaCl treatment?
Why do the graphs in for example Fig 4B look different from the graphs in Fig 5. Use the same formatting across the whole paper
Fig 8B graphs – x-axis “Control” and y-axis “Content”– fix spelling mistakes.
Fix the use of in-text references, that the publication year is indicated in italics and sometimes not.

Annotated reviews are not available for download in order to protect the identity of reviewers who chose to remain anonymous.

·

Basic reporting

The manuscript can be accepted for publication.

Experimental design

I have no concerns about the experimental design.

Validity of the findings

The findings presented in this study is suitable for publication in PeerJ.

---

## Round 0.3 · Minor Revisions

Thank you for your response. Regarding the letters which highlight the existence of significant differences, it is highly advisable to not repeat the same letter with a different meaning in the same graph (e.g. one should not use the letter "a" in the controls and the same letter in the "experimental" bars unless one intends to state that the experimental bar labelled "a" is not significantly different from the control bar labelled "a") In your figures, you have repeated the same letter with different meanings in figs 4 and 6, whereas in figures 2,5,7, and 9 the letters have consisten meaning (i.e. they clearly refor to comparisons between all data) and in figure 3 and 8 I cannot tell which convention is being used. Please recheck your figures and relabel them using letters that refer to comparisons between all bars, since otherwise reading and interpreting your figures becomes quite confusing and probably detracts from your message. The inclusion of a description of the different meaning in Figure 4 is not very helpful, since one expects all figures to use the same convention and the reader mahy unwittingly not notice that.

---

## Round 0.4 · accepted · Accept

I am glad to accept your paper.